# Genomic Mutations of BK Polyomavirus in Patients after Kidney Transplantation: A Cross-Sectional Study in Vietnam

**DOI:** 10.3390/jcm11092544

**Published:** 2022-05-01

**Authors:** Truong Quy Kien, Pham Quoc Toan, Phan Ba Nghia, Diem Thi Van, Nguyen Van Duc, Do Manh Ha, Nguyen Thi Thuy Dung, Nguyen Thi Thu Ha, Le Thi Bao Quyen, Hoang Trung Vinh, Bui Van Manh, Hoang Xuan Su, Tran Viet Tien, Le Viet Thang, Lionel Rostaing

**Affiliations:** 1Department of Nephrology, Military Hospital 103, Vietnam Military Medical University, Hanoi 100000, Vietnam; drquykientruong@gmail.com (T.Q.K.); toannephro@gmail.com (P.Q.T.); phannghiaba@gmail.com (P.B.N.); diemvan1989@gmail.com (D.T.V.); nguyenvanducyhhk@gmail.com (N.V.D.); manhhado215@gmail.com (D.M.H.); bsdunga12@gmail.com (N.T.T.D.); drthuha103@gmail.com (N.T.T.H.); hoangvinh.hvqy@gmail.com (H.T.V.); lethangviet@yahoo.co.uk (L.V.T.); 2Department of Microbiology, Faculty of Biology, University of Science, National University of Hanoi, Hanoi 100000, Vietnam; baoquyen271295@gmail.com; 3Center of Emergency, Intensive Care Medicine and Clinical Toxicology, Hanoi 100000, Vietnam; drmanhbui@gmail.com; 4Institute of Biomedicine and Pharmacy, Vietnam Military Medical University, Hanoi 100000, Vietnam; hoangxuansu@vmmu.edu.vn; 5Department of Infectious Disease, Vietnam Military Medical University, Hanoi 100000, Vietnam; tientv@vmmu.edu.vn; 6Nephrology, Hemodialysis, Apheresis and Kidney Transplantation Department, Grenoble University Hospital, 38000 Grenoble, France

**Keywords:** BK polyomavirus, VP1 region, single nucleotide polymorphism, genotype, amino acid, kidney transplantation, Vietnam

## Abstract

Objectives: The purpose of this study was to identify the SNP sites and determine the BKV genotype circulating in kidney-transplant Vietnamese recipients based on the VP1 gene region. Methods: 344 samples were collected from post-kidney-transplant recipients at the 103 Vietnam Military Hospital to investigate the number of BKV infections. Positive samples with a sufficient virus concentration were analyzed by nested PCR in the VP1 region, sequencing detected genotyping and single-nucleotide polymorphism. Results: BKV infection was determined in 214 patients (62.2%), of whom 11 (5.1%) were diagnosed with BKV-associated nephropathy. Among the 90 BKV-I strains sequenced, 89 (98.88%) were strains of I/b-1 and 1 (1.12%) was strain I/b-2. The 60 BKV-IV strains had a greater diversity of subgroups, including 40% IV/a-1, 1.66% IV/a-2, 56.68% IV/c-1, and 1.16% IV/c-2. Additionally, of 11 cases diagnosed with BKVN, seven belonged to subgroup I/b-1 (63.6%) and four to subgroup IV/c-1 (36.4%). Moreover, 22 specific SNPs that were genotype I or IV were determined in this Vietnamese population. Specifically, at position 1745, for the Vietnamese BKV-IV strains, the SNP position (A→G) appeared in 57/60 samples (95%). This causes transformation of the amino acid N→S. This SNP site can enable detection of genotype IV in Vietnam. It represents a unique evolution pattern and mutation that has not been found in other international strains. Conclusion: The BKV-I genotype was more common than BKV-IV; however, mutations that occur on the VP1 typing region of BKV-IV strains were more frequent than in BKV-I strains.

## 1. Introduction

The human BK polyomavirus (BKV) belongs to the Polyomaviridae family, which is characterized by a double-stranded circular DNA genome that has been isolated from several hosts, including humans, monkeys, rabbits, rodents, and birds [1]. Its genome is approximately 5.3 kb and contains an early region, a late region, and a non-coding control region (NCCR) [2]. BKV is often acquired during early childhood with a seroprevalence of 60–80% in the general population [3]. Following a primary infection, the virus remains latent and mainly asymptomatic within the kidney and urinary tract [4]. It can become active during severe immunosuppressive therapy due to kidney or stem-cell transplantation: reactivation of BKV can then occur, and then replicates massively, resulting in damage to uroepithelial cells [2,5]. 

In kidney-transplant recipients, the prevalence of BKV reactivation ranges from 10–60%, of which 1–10% develop BKV nephropathy (BKVN), characterized by severe tubulointerstitial nephritis and basement-membrane necrosis [6,7]. Progression of BKVN usually occurs without any clinical signs or symptoms, except for rising serum-creatinine levels. Unless detected promptly, 50% of BKVN cases can cause graft loss within the following months/years [3].

The major capsid component, VP1, can be divided into five outer domains: i.e., BC, DE, EF, GH, and HI, which connect the various β-strands and α-helix of the polypeptide segments [7]. The BC and EF loop regions are most frequently affected by mutations and the BC loop contains a short sequence at positions (1744–1812) used to identify the four main viral genotypes [7,8]. BKV has been classified into genotypes and subgroups based on single-nucleotide polymorphism (SNPs) analysis of the VP1 region and the non-coding control region (NCCR) [9]. Nucleotides of the VP1 coding region show very high conservation (over 95%) in all genotypes of BKV, but the similarity between the amino acid residues from 61E to 83R is only 61–70% [7,10].

Genotype I is the most prevalent and widespread worldwide (about 80% of reported cases), followed by genotype IV (about 15% of reported cases), mainly distributed in Europe and East Asia, while genotypes II and III are rare in all geographic regions (~5%). Genotypes I and IV are further divided into four subgroups of I (I-a, I-b1, I-b2, and I-c) and six of IV (IV-a1, IV-a2, IV-b1, IV-b2, IV-c1, and IV-c2) [11]. Analysis of polymorphism between obtained sequences, especially in the BC loop, is vital in medical research because molecular variation in BKV may entail changes in tropism and influence the clinical manifestations of infection [12]. However, in-depth studies on the molecular genetics of BKV have rarely been reported in Vietnam. Therefore, the purpose of this study was to identify the SNP sites and to determine the BKV genotype circulating in kidney-transplant recipients in Vietnam, based on the VP1 gene region.

## 2. Materials and Methods

### 2.1. Sample and DNA Extraction

A total of 344 samples were collected from post-kidney-transplant recipients at the 103 Vietnam Military Hospital. We investigated the number of BKV infections using qPCR. Among the 344 samples, 150 urine or serum samples were selected from patients diagnosed with BKV. Viral DNA was extracted from 200 μL of clinical sample (plasma or urine) using a Geneall DNA viral kit (Geneall, Seoul, Korea), according to the manufacturer’s protocol. The DNA was finally eluted in a final volume of 54 μL of AE buffer and stored at −80 °C until use.

### 2.2. Quantification of BK Viral Load in Urine and Plasma Samples

We used an in-house quantitative realtime PCR assay developed in our laboratory with a slight modification from a previously published protocol on the Rotor Gene Q5 plex MDx platform (Qiagen, Germany) for quantification of BK virus load in urine and plasma [13]. The BK virus load was expressed in BKV genome copies per milliliter of urine or plasma. The lower limit of BK viral load detection at our center is 250 copies per milliliter.

### 2.3. Genotyping of BKV

Primers were designed for the VP1 region of BKV, as described previously [14,15]. The external primer pair, BKS + BKAS, and the internal primer pair, BKF + BKR1, amplified a 580-bp and a 327-bp DNA fragment, respectively (Table 1). The PCR reaction was optimized in a total volume of 20 μL using a PCR kit (GoTaq Mastermix, Promega, Madison, WI, USA) containing 5 μL of DNA template, 1 μL primer, 4 μL of molecular-grade water, and 10 μL of 2X Buffer. Cycling conditions were 95 °C for 5 min, followed by 35 cycles of 94 °C for 30 s, 58 °C for 30 s, and 72 °C for 30 s, and a final extension at 72 °C for 7 min. For the nested PCR, 3 µL of the products from the PCR, using the external primer pair, was added as the DNA template for the second-round of PCR using the internal primer pair. PCR amplifications were performed using the parameters described above. All reactions were implemented on an Eppendorf™ Mastercycler™ pro PCR System. Amplification products were separated by electrophoresis on 1.2% agarose and visualized under UV light after staining with ethidium bromide (10 mg/mL) and sequenced using a 3130xl sequencer.

### 2.4. Phylogenetic Analysis

All of the obtained sequences were compared with 31 reference sequences retrieved from GenBank using Bioedit 7.0 (http://www.mbio.ncsu.edu/bioedit/bioedit.html, Hall 1999 (accessed on 10 March 2022)) and MEGA 7.0 software (https://www.megasoftware.net/ (accessed on 10 March 2022)) (Table 2). The data were then analyzed by constructing a phylogenetic tree using the neighbor-joining method, and significance level was estimated with 1000 bootstrap replicates.

The basis of specific polymorphisms in the portion of the VP1 region, spanning nucleotide (1650–1955) was determined and analyzed in terms of the changes in one or more amino-acid sequences using Bioedit 7 (http://www.mbio.ncsu.edu/bioedit/bioedit.html, Hall 1999 (accessed on 10 March 2022)).

### 2.5. Renal Allograft Biopsy

Allograft biopsy was performed in patients with acute allograft dysfunction, BK virus nephropathy was characterized by tubular atrophy and fibrosis with a variable inflammatory lymphocytic infiltrate, BK virus nephropathy was confirmed using immunohistochemical nuclear staining with anti-SV40 antibody.

### 2.6. Statistical Analysis

Differences between categorical variables were analyzed by the χ^2^ test while independent t-test or Mann-Whitney test were used to compare quantitative variables. All statistical analyses were conducted using the SPSS software version 20.0 (IBM, Armonk, NY, USA). *p* value less than 0.05 was considered to be statistically significant.

## 3. Results

### 3.1. Characteristics of the Kidney-Transplant Recipients

Of the 344 kidney-transplant recipients, BKV was detected in 214 (62.2%), of which 11 (5.1%) developed BKV nephropathy. Among the 214 positive-BKV recipients, 24 (7.0%) only developed BKV viremia, and 50 had BK viruria alone was present (6.98%), whereas 140 (14.5%) had both BKV viremia and BKV viruria. The median values for BKV viremia and viruria were 3.95 ± 1.14 and 6.81 ± 2.82 log copies/mL, respectively (Table 3).

We also analyzed for possible associations between the two groups infected with BKV and without BKV. The results showed that there was no significant difference between the two groups with regards to age, gender, HLA mismatch, or types of immunosuppressive drugs. However, the difference between the BKV-negative group and BKV-positive group in serum-creatinine concentration (115.95 ± 36.84 vs. 101.25 ± 23.19, respectively) and estimated glomerular-filtration rate (58.95 ± 14.60 vs. 65.12 ± 14.19, respectively) were determined to be statistically significant (*p* < 0.001, Table 4).

To investigate the possible impacts of the different genotypes, we also compared the differences of patients infected with BKV-I and BKV-IV. However, there were no statistically significant differences in age, sex, HLA mismatch, time after transplantation, serum creatinine, eGFR, BKVN, immunosuppressive therapy, and BKV load between the two groups (Table 5).

### 3.2. Phylogenetic Analysis of BKV Isolates 

A total of 150 samples were genotyped by constructing a phylogenetic tree from analysis of 305-bp fragments of BKV, aligned with 31 reference sequences retrieved from the GenBank (Figure 1 and Figure 2). The results indicated that 90 (76.6%) belonged to BKV-I and 60 (23.3%) to BKV-IV, with no cases of BKV-II or BKV-III observed. Among the 90 BKV-I strains, 89 (98.9%) strains were of I/b-1 and 1 (1.1%) of I/b-2. BKV-IV had a greater diversity of subgroups, including 40% of IV/a-1, 1.7% of IV/a-2, 56.7% of IV/c-1, and 1.2% of IV/c-2 (Table 6). Of the 11 cases that were diagnosed with BKVN, seven were subgroup I/b-1 (63.6%) and four were subgroup IV/c-1 (36.4%).

The 305-bp typing region sequenced in this study, along with 31 reference sequences, were used to construct a neighbor-joining phylogenetic tree using Kimura’s correction.

### 3.3. Analysis of SNP

Analysis of 31 reference sequences and 150 sequences that contained the VP1 gene fragment of 305 bp in size (1650–1955) showed 22 different SNP positions between genotypes I and IV (Table 7). Among these, 10 had capsid amino-acid sequence substitution of BKV (61, 62, 66, 69, 71, 74, 75, 77, 82, 117).

Ninety of the BKV-I sequences were further compared with Dunlop strains and international strains of subgroup I in the VP1 gene region (1650–1955) to determine polymorphisms. Sequence analysis showed that 15 SNPs were detected within subgroup I; however, most of these SNP sites appeared singly, except at nucleotide position 1786, as hot spots (Table 8). At this position there was a nucleotide change (G → A) in the amino acid substitution, i.e., Asp → Asn, in a frequency of 9/90 samples (10%).

Sixty BKV-IV sequences were compared with international strains of subgroups IV/a-1 (AB269869), IV/a-2 (AB211389), IV/c-1 (AB269867), and the parent strain (sequence IV, Z19535, Jin L (1993) [3], isolated in the UK) (Table 9). After analysis, there were 15 distinct SNP positions between parent strain IV and isolated strains found in Vietnam, in which many SNPs had changed amino acids. It is notable that most of these SNP sites appeared singly, but all had an altered amino acid sequence (Table 9 and Table 10). Specifically, seven SNP positions that belong to subgroup IV had high frequency, of which 1745 and 1794 had changed amino acid-sequences (95.0% and 61.7%, respectively).

## 4. Discussion

We reported on the prevalence of BKV infection in the blood and urine of 344 kidney-transplant recipients, of which 214 were BKV-positive (62.2%), which was a slightly lower rate than the study of Toan et al. in northern Vietnam (77.4%) [16]. This difference may be due to our larger sample size (344 in our study vs. 82 for Toan et al.’s study). However, the number of cases of BK viruria and BK viremia in kidney-transplant recipients in our study is similar to that reported in other studies [17,18].

Previous studies have reported a number of risk factors for BKV nephropathy, including age, gender, HLA mismatches, deceased-donor transplants, and types of immunosuppressive drugs. Nevertheless, in our study, there was no difference between groups with and without BKV infection regarding any of the risk factors mentioned, except for differences in serum-creatinine and estimated glomerular-filtration rate, which were statistically significant (*p* < 0.001, Table 4). As BKV infection represents a threat to premature allograft loss and the early monitoring of BKV load using realtime PCR in either plasma or urine can improve the management of patients after kidney transplantation.

The first genotyping scheme for BKV was reported by Jin et al. in 1993 using restriction fragment-length polymorphism (RFLP) based on nucleotide polymorphisms in a very short fragment of the gene of the capsid protein VP1 (nucleotides 1744 to 1812). Sanger sequencing is now a major method of genotyping BKV [19]. In previous studies, the phylogenetic tree was constructed based on a complete genome sequence or any viral genomic region; however, the VP1 gene region contains more polymorphism information that can yield a clearer definition of subgroups [6,20].

BKV can be classified into four genotypes (I → IV) and each genotype is further divided into several subgroups according to geographical distribution [21]. BKV-I was the predominant genotype during the period of study follow at 60% overall (90 of 150 BKV-positive cases), followed by BKV-IV (60 of 150 BKV-positive cases), and no cases of genotypes II or III were found. Intra-genotype diversity was observed in both genotypes; however, BKV-IV revealed more subgroup variety than BKV-I. In genotype I, all sequences belonged to Ib1 except one, which belonged to subgroup Ib2. Subtype Ib-1 has been shown to be highly prevalent in southeast Asia, across Europe, and parts of North Africa, while subtype Ib-2 is common in Europe [16,22].

The only patient in this study with subgroup Ib-2 BKV received a kidney transplant in France. This can be explained because that patient had received a kidney from a donor with BK viruria. Recent studies have concluded that BKV replication in recipients come from donor origin [23,24]. However, the coexisting four subgroups of IV (IV-a1, IV-a2, IV-c1, and IV-c2), and the three subgroups (IV-a1, IV-a2, IV-c1), are the same as reported in previous publications [16,25]. Interestingly, one patient of this study belonged to IV-c2 which was prevalent in Mongolia and Europe [25]. In this situation, the patient or kidney donor could have acquired the BKV infection not from living local community but from geographical regions where BKV-IV/c2 was predominant, which supported the source of BKV after kidney transplantation. 

Nukuzuma.S et al. (2006) [26] showed that BKV-I replicates more efficiently than BKV-IV in human renal epithelial cells, while several other reports illustrated the association between genotypes BKV-II, III, IV with a higher risk of BKVN in kidney transplant patients [27,28]. Therefore, the role of particular genotypes or BKV variants in the development of BKVN remains unclear. In this study, we compared two groups of patients, BKV-I and BKV-IV, with some clinical factors. In general, the proportion of patients infected with BKV-I going on to develop BKVN was higher than patients with BKV genotype IV. This can be explained because BKV-I dominates the population of Vietnam as well as by the limited number of BKVN cases in studies. Therefore, a large cohort of BKVN cases should be analyzed to determine the role of BKV genotypes on the pathogenesis and the clinical course of BKVN in patients with a history of kidney transplant.

Of note, suitably selected SNP assays can provide an alternative way for BKV genotyping. A recent study constructed an algorithm in order to determine the 12 BKV subgroups and subtypes based on a region comprising only 100 bp of the VP1 gene (1977 through 2076) with 12 isolated SNPs [29]. Twenty-two specific SNPs for the identification of genotype I or IV in a Vietnamese population were determined (Table 7), of which 12 sites (1760, 1787, 1793, 1848, 1912, 1704, 1760, 1770, 1784,1854, 1809, and 1938) concordant with the results reported by Luo et al. [20]. A subgroup Ib2 sequence has two SNPs (1687 and 1908) showing the remaining known subtype Ib1 strains, but there is no special SNP position to separate the IV subgroups from each other on the VP1 region (1650–1955). Therefore, a study that investigated SNP sites to differentiate subgroups IV should be performed on other genomic regions of BKV, such as LTA or NCCR, because the application of SNP genotyping can remove the dependence on the accuracy of the algorithm when creating a phylogenetic tree. 

It is notable that single nucleotide polymorphisms (SNPs) are believed to be vital in the pathogenesis of the virus as well as a mechanism of immune escape. Varella et al. (2017) found differences in viral load between the Ia and Ib1 subgroups, which discriminate in the sequences coding for VP1 [30]. Therefore, changes in the single nucleotide sequence can affect not only its pathogenicity, but also the tropism of the virus or the range of hosts. At position 1786 (corresponding to amino acid 75) of genotype I on some Vietnamese strains, the SNP position (G → A) appeared to change the amino acid Asp → Asn, observed in 9/90 samples (10%). This is a rare SNP site: not within the range of SNPs that Jin et al. described in 1993 to discriminate BKV genotypes [25], but recognized as a rare mutation by Tremolada et al. (2010) and was often present in the urine in patients with BKVN [2]. Most genotype I SNP sites appear singly, whereas, for genotype IV, we observed that 7/15 SNP sites (1737, 1745, 1794, 1851, 1860, 1905, 1938) had high repetition frequencies (28–95%). In particular, position 1745 on Vietnamese strains shows the SNP position (A → G) in 57/60 samples (95%). This causes an amino acid substitution from N → S. To our knowledge, this is a distinct SNP site and can be used for detection of genotype IV in Vietnam, representing a unique evolution pattern and mutation that has not been found in other international strains. However, limitations also exist, because our samples were collected from a particular geographical area. Additional sequences from other areas in Vietnam, including central and south Vietnam, are required to confirm these results. Additionally, this is only a cross-sectional study at a time, therefore the assessment of some related factors has not been really rigorous.

## 5. Conclusions

The BKV-I genotype was more common than BKV-IV; however, mutations that occur on the VP1 typing region of BKV-IV strains were more frequent than in BKV-I strains. This study has provided more insights into the genotype characteristics and SNP sites of BKV strains among recipients of renal transplant in Vietnam.

## Figures and Tables

**Figure 1 jcm-11-02544-f001:**
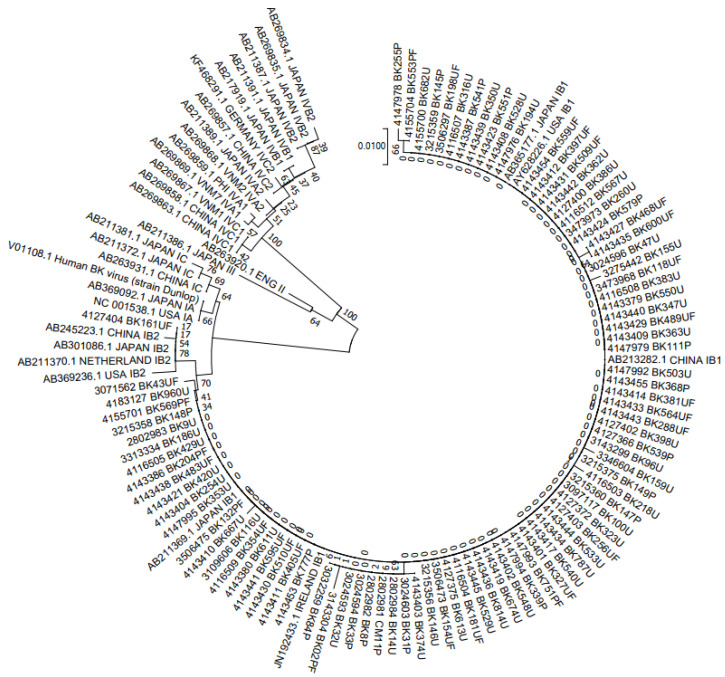
NJ phylogenetic tree clustering of 90 BKV-I polyomavirus sequences to determine subgroups.

**Figure 2 jcm-11-02544-f002:**
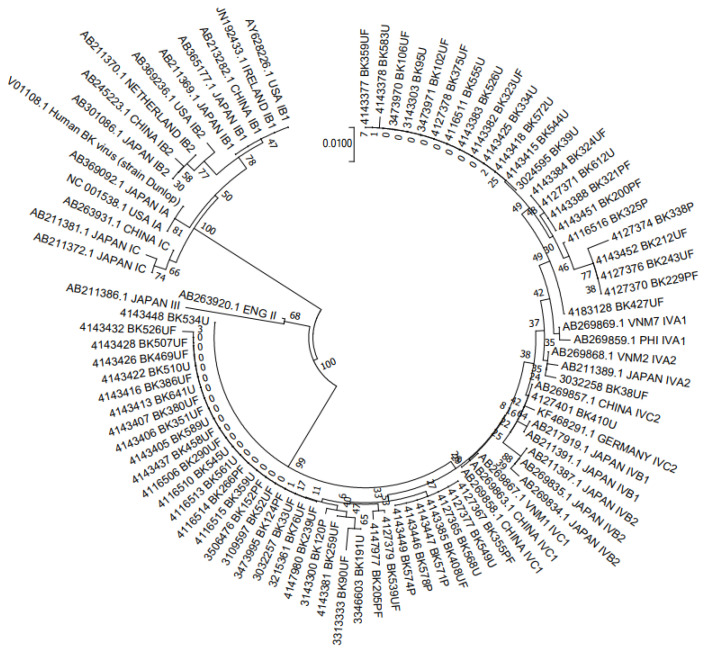
NJ phylogenetic tree clustering 60 BKV-IV polyomavirus sequences to determine subgroups.

**Table 1 jcm-11-02544-t001:** Primer sequences used in this study.

Primers	Sequences	Position *	Size	References
BKV_S	ATC AAA GAA CTG CTC CTC AAT	1361–1381	580 bp	[15]
BKV_AS	GCA CTC CCT GCA TTT CCA AGGG	1919–1940		
BKV-F	CAA GTG CCA AAA CTA CTA AT	1630–1649	327 bp	[14]
BKV-R1	TGC ATG AAG GTT AAG CAT GC	1937–1956		

Abbreviations: BKV, BK polyomavirus; bp, base pair. * With respect to Genebank accession number V01108 (Dunlop strain).

**Table 2 jcm-11-02544-t002:** Reference sequence.

Access No.	Subgroups	Country
AB211369	Ib1	Japan
AB365177	Ib1	Japan
AB213282	Ib1	China
JN192433	Ib1	Ireland
AY628226	Ib1	USA
AB369236	Ib2	USA
AB301086	Ib2	Japan
AB245223	Ib2	China
AB211370	Ib2	Netherland
AB369092	Ia	Japan
NC001538	Ia	USA
V01108	Ia	USA
AB263931	Ic	China
AB211381	Ic	Japan
AB211372	Ic	Japan
AB211386	III	Japan
AB263920	II	English
AB269859	IV a1	Philippine
AB269869	IV a1	Vietnam
AB268868	IV a2	Vietnam
AB211389	IV a2	Japan
AB269867	IV c1	Vietnam
AB269858	IV c1	China
AB269863	IV c1	China
AB211391	IV b1	Japan
AB217919	IV b1	Japan
KF468291	IV c2	Germany
AB269857	IV c2	China
AB269835	IV b2	Japan
AB269834	IV b2	Japan
AB211387	IV b2	Japan

**Table 3 jcm-11-02544-t003:** Characteristics of kidney-transplant recipients.

Variable	Value
Age (years) (mean ± SD)	38.37 ± 10.09
Gender	
Male, *n*, %	245 (71.2%)
Female, *n*, %	99 (28.8%)
Times of HD (months)	28.26 ± 41.70
HLA mismatch (mean ± SD)	3.36 ± 1.18
Serum creatinine at first BKV testing (µmol/L)	110.40 ± 33.10
eGFR at first BKV testing (mL/min)	61.28 ± 14.75
BK viremia only, *n*, %	24 (7.0%)
BK viruria only, *n*, %	50 (14.5%)
BK viremia and viruria, *n*, %	140 (40.7%)
BKV, *n*, %	214 (62.2%)
BK viremia (log copies/mL)	3.95 ± 1.14
BK viruria (log copies/mL)	6.81 ± 2.82
CNIs	
Tacrolimus, *n* (%)	305 (88.66%)
Cyclosporine, *n* (%)	39 (11.34%)

Abbreviations: HD, hemodialysis; BKV, BK virus; SD, standard deviation; eGFR, estimated glomerular filtration rate; CNI, calcineurin inhibitor.

**Table 4 jcm-11-02544-t004:** Comparison between BKV and non-BKV groups.

Variable	BKV (−)	BKV (+)	*p* Value
Age (years) (mean ± SD)	39.01 ± 9.99	37.98 ± 10.15	0.36
Gender			0.22
Male, *n*, %	95 (73.1)	150 (70.1)	
Female, *n*, %	35 (26.9)	64 (29.9)	
HLA mismatch (mean ± SD)	3.33 ± 1.02	3.37 ± 1.27	0.75
Serum creatinine at first BKV testing (µmol/L)	101.25 ± 23.19	115.95 ± 36.84	<0.001
eGFR at first BKV testing (mL/min)	65.12 ± 14.19	58.95 ± 14.60	<0.001
Times after kidney transplantation (months, mean ± SD)	5.73 ± 5.70	10.79 ± 18.40	0.003
CNIs			
Tacrolimus, *n*, %	118 (90.7)	187 (87.4)	0.62
Cyclosporine, *n*, %	12 (9.3)	27 (12.6)	

Abbreviations: SD, standard deviation; eGFR, estimated glomerular filtration rate; CNI, calcineurin inhibitor; BKV, BK virus.

**Table 5 jcm-11-02544-t005:** Comparison between BKV-I and BKV-IV groups.

Variable	BKV-I	BKV-IV	*p* Value
Age (years) (mean ± SD)	38.44 ± 9.67	35.98 ± 10.69	0.15
Gender			0.69
Male, *n*, %	61 (67.8)	44 (73.3)	
Female, *n*, %	29 (32.2)	16 (26.7)	
HLA mismatch (mean ± SD)	3.40 ± 1.35	3.43 ± 1.20	0.9
Serum creatinine at first BKV testing (µmol/L)	113.96 ± 35.40	126.42 ± 46.86	0.07
eGFR at first BKV testing (mL/min)	59.07 ± 15.26	57.03 ± 16.81	0.44
log BK viremia (mean ± SD)	3.95 ± 1.13	4.28 ± 1.19	0.11
log BK viruria (mean ± SD)	7.54 ± 2.65	7.19 ± 2.90	0.45
BKVN *n*, %	7 (7.8%)	4 (6.7%)	0.81
CNIs			
Tacrolimus, *n*, %	78 (86.7)	52 (86.7)	0.59
Cyclosporine, *n*, %	12 (13.3)	8 (13.3)	

Abbreviations: SD, standard deviation; eGFR, estimated glomerular filtration rate; CNI, calcineurin inhibitor; BKV, BK virus.

**Table 6 jcm-11-02544-t006:** Distribution of BKV subgroups.

BKV-I (*n* = 90)				BKV-IV (*n* = 60)				
I/a	I/b-1	I/b-2	I/c	IV/a-1	IV/a-2	IV/b-1	IV/c-1	IV/c-2
0 (0%)	89 (98.9%)	1 (1.12%)	0 (0%)	24 (40%)	1 (1.7%)	0 (0%)	34 (56.7%)	1 (1.7%)

**Table 7 jcm-11-02544-t007:** SNP difference between BKV-I and BKV-IV.

SNP Position	Aa Position	BKV-I	BKV-IV	Aa Substitution
1704	47	G	A	
1716	51	C	T	
1722	53	C	T	
1744	61	G	A	E → N/S
1746	61	A	T	
1747	62	A	G	N → D
1760	66	T	A	F → Y
1769	69	A	G/A (3)	K → R
1770	69	G	A	
1775	71	G	C	S → T
1784	74	A	C	N → T
1787	75	A	C	D → A
1792	77	A	G/C	S → D/N/E/Q/H
1793	77	G	A	
1809	82	G/A	C	E → D
1848	95	C	A A	
1851	96	C	A/G	
1854	97	C	T	
1869	102	C	T	
1890	109	G	A	
1912	117	C	A	Q → K
1938	125	C	T	

**Table 8 jcm-11-02544-t008:** SNP positions among BKV-I subgroups in the VP1 gene region.

SNP	Dunlop (I/a)	AB211369 (I/b-1)	AB211370 (I/b-2)	AB211372 (I/c)	Research Sample	Frequency
1687	G	G	C	G	G/C	89/90, 1/90
1698	T	A	A	A	A	90/90
1741	G	G	G	G	G/C	88/90, 2/90
1765	C	G	G	G	C/G	89/90, 1/90
1780	G	G	G	G	G/C	89/90, 1/90
1786	G	G	G	G	A/G	9/90, 81/90
1809	G	A	A	G	A	90/90
1848	C	C	C	C	C/T	89/90, 1/90
1863	T	T	T	T	T/C	88/90, 2/90
1908	T	T	A	T	T/A	89/90, 1/90
1919	A	A	A	A	A/G	89/90, 1/90
1923	T	C	C	T	C	90/90
1929	A	A	A	A	A/G	89/90, 1/90
1946	A	A	A	A	A/C	89/90, 1/90
1952–1953	AT	AT	AT	AT	AT/TG	89/90, 1/90

**Table 9 jcm-11-02544-t009:** SNP positions among BKV-IV subgroups in the VP1 gene region.

SNP	IV	IV/a-1	IV/a-2	IV/c-1	IV/c-2	Research Sample	Frequency
1735	G	G	G	G	G	G/A	55/60, 5/60
1737	T	T	T	T	T	T/C	38/60, 23/60
1741	G	G	G	G	G	G/C	59/60, 1/60
1745	A	A	A	A	A	A/G	3/60, 57/60
1769	G	G	G	G	G	G/A	57/60, 3/60
1780	G	G	G	G	G	G/T/C	57/60, 1/60, 2/60
1781	A	A	A	A	A	A/C	59/60, 1/60
1786	G	G	G	G	G	G/A	59/60, 1/60
1792	G	G	G	G	G	G/A/C	51/60, 3/60, 6/60
1794	G	G	C	C	C	C/G	37/60, 23/60
1851	A	A	A	A	A	A/G	38/60, 22/60
1860	G	G	A	G	G	A/G	24/60, 36/60
1870	C	C	C	C	C	C/T	59/60, 1/60
1905	G	G	G	A	G	A/G	35/60, 25/60
1938	C	T	T	T	T	T/C	43/60, 17/60

**Table 10 jcm-11-02544-t010:** Amino acid substitution between strain IV, Dunlop, and the study strains.

SNP	AA Position	Dunlop AA	IV AA	Research AA
1735	58	D	D	D(55/60), N(5/60)
1737	58	D	D	
1741	60	D	D	D (59/60), H (1/60)
1745	61	E	N	N (3/60), S (57/60)
1769/1770	69	K	R	R (57/60), K (3/60)
1780/1781	73	E	S	E (57/60), Q (2/60), S (1/60)
1786	75	D	A	A (59/60), T (1/60)
1792	77	S	E	D (33/60), E (18/60), Q (5/60), H (1/60), N (3/60)
1794	77	S	E	
1851	96	L	L	L (60/60)
1860	99	L	L	L (60/60)
1870	103	L	L	L (60/60)
1905	114	V	V	V (60/60)
1938	125	S	S	S (60/60)

Abbreviations: AA, aminoacid.

## Data Availability

Data are available upon reasonable request.

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
