# Peer review of "Genomic Mutations of BK Polyomavirus in Patients after Kidney Transplantation: A Cross-Sectional Study in Vietnam"

_jcm, 2022, doi:10.3390/jcm11092544_

Round 1

Reviewer 1 Report

The manuscript had provided insight into the genotype and SNP sites of BKPyV strains in Vietnam. And they found a special SNP site that could enable detection of genotype IV in Vietnam. However, the VP1 genotype has been studied in detail. Therefore, the results are not really novel from my perspective. But this study really could provide large sample for BKPyV study. More specific comments include:

Major:

  1. Please provide the detailed diagnosis for viruria, viremia and BKPyVN. Different diagnosis might cause different results based on the previous studies. In addition, the detailed diagnosis might help to compare this study to others.
  2. Please provide the sequences of the study. Why did you select the “neighbor-joining method” to do the analysis? Different methods might lead to different results.
  3. On line 212-219, only one sample could not support the conclusion. In fact, the NCCR is the highest mutant region in the BKPyV genome, which was also related to the progression of BKPyVN.

Minor:

  1. The reactivate of BKPyV is still unclear, only one sample could not support the BKPyV came from donor.
  2. Please provide the explain about why ‘age, gender, HLA mismatches, deceased-donor transplants, and types of immu-189 nosuppressive drugs’ were not different between positive and negative BKPyV groups.
  3. It is noted that your manuscript needs careful editing by someone with expertise in technical English editing paying particular attention to English grammar, spelling, and sentence structure so that the goals and results of the study are clear to the reader.

Author Response

Major corrections.

Comment 1. Please provide the detailed diagnosis for viruria, viremia and BKPyVN. Different diagnosis might cause different results based on the previous studies. In addition, the detailed diagnosis might help to compare this study to others.

Response 1: We agree with your comments and we added the detailed diagnosis for viruria, viremia and BKPyVN in 2.2 section on materials and methods.

Comment 2. Please provide the sequences of the study. Why did you select the “neighbor-joining method” to do the analysis? Different methods might lead to different results.

Response 2: We added information about the reference sequences which were presented in Table 2 to distinguish the study sequence and the reference sequence in the phylogenetic tree. We also provide more details of the primer sequences used in the study (Table 1).

Neighbor-joining method is a distance-based method which is relatively simple and direct. The distance (roughly, the percent sequence difference), is calculated for all pairwise combinations of operation taxonomic units, and then the distances are gathered into a tree. Besides, NJ is a fast method suited for large datasets and permits different branch lengths indicating the evolutionary time or amounts of evolutionary changes along the branch. Additionally, this is also the most chosen method to construct phylogenetic tree on VP1 gene region of BKV from previous studies.

Comment 3. On line 212-219, only one sample could not support the conclusion. In fact, the NCCR is the highest mutant region in the BKPyV genome, which was also related to the progression of BKPyVN.

Response 3: We didn’t use NCCR for genotyping because of the following reasons: (i) the presence of insertions, deletions, and rearrangements make accurate alignment in this region difficult, (ii) it is recognized that rearrangement patterns in the NCCR do not correlate with conventional viral genotypes, and (iii) when large indels occur, one cannot decide if they represent single or multiple mutational events.

Actually, we can still accurately distinguish subgroups based on VP1 region in this study. We have also revised the content of this paragraph to better match the conclusion as follow “The only patient in this study with subgroup Ib-2 BKV received a kidney transplant in France. This can be explained that patient had received a kidney from a donor with BK viruria, as reported in recent studies concluded that BKV replication in recipients come from donor origin [23], [24]. However, the coexisting four subgroups of IV (IV-a1, IV-a2, IV-c1, and IV-c2), and the three subgroups (IV-a1, IV-a2, IV-c1), are the same as reported in previous publications [16], [25]. Interestingly, one patient of this study belonged to IV-c2 which was prevalent in Mongolia and Europe [25]. In this situation, the patient or kidney donor could acquire BKV infection not from living local community but from geographical regions where BKV-IV/c2 was predominant, supported the source of BKV after kidney transplantation”.

Comment 4. It is noted that your manuscript needs careful editing by someone with expertise in technical English editing paying particular attention to English grammar, spelling, and sentence structure so that the goals and results of the study are clear to the reader.

Response 4: Thank you for your valuable comment. We have revised the manuscript thoroughly according to your suggestion.

Minor points

Comment 1: The reactivate of BKPyV is still unclear, only one sample could not support the BKPyV came from donor

Response 1: Thank you for your valuable comment. We have revised this paragraph on page 10, lines 238-246, as follows. The only patient in this study with subgroup Ib-2 BKV received a kidney transplant in France. This can be explained that patient had received a kidney from a donor with BK viruria, as reported in recent studies concluded that BKV replication in recipients come from donor origin [23], [24]. However, the coexisting four subgroups of IV (IV-a1, IV-a2, IV-c1, and IV-c2), and the three subgroups (IV-a1, IV-a2, IV-c1), are the same as reported in previous publications [16], [25]. Interestingly, one patient of this study belonged to IV-c2 which was prevalent in Mongolia and Europe [25]. In this situation, the patient or kidney donor could acquire BKV infection not from living local community but from geographical regions where BKV-IV/c2 was predominant, supported the source of BKV after kidney transplantation.  

Comment 2: Please provide the explain about why ‘age, gender, HLA mismatches, deceased-donor transplants, and types of immu-189 nosuppressive drugs’ were not different between positive and negative BKPyV groups.

Response 2: This can be explained that our study is a cross-sectional study, including the different post-transplant time of patients and the majority of living organ donors. Additionally, there are some limitations in the selection of immunosuppressive regimens and experience in the treatment of transplant rejection leading to the majority of kidney recipients have high HLA compatibility and low pre-sensitization rate (PRA). Therefore, it may affect the results of the study to some extent.

Reviewer 2 Report

The goal of this essay was to characterize SNPs in BKV sequences collected from tx patients in a cohort of patients in Vietnam. The work is methodologically well conducted for its purpose and includes a good number of samples which enriches the analytical potential of the results. Genotypic investigation has also interesting findings. However, being a microbiology journal of clinical interest, some findings could have been better explored in order to identify possible impacts of these in the cohort of patients. Here are some notes that I think are relevant:

1) Methodology: the authors don´t mention BK loads quantification but present in results. Please add viral load procedures in methods.

What post-transplant stage are the patients in? 1 year? 6 months? What is the range?

2) Results: I missed an investigation into the possible impacts of the different genotypes and the mutations observed on variables such as viral load, detection of viremia and viruria, increase in creatinine, glomerular filtration and nephropathy, which would have been interesting to explore.

3) Discussion: the cross-sectional nature of the investigation should, in addition to those presented above, have been also mentioned as an important limitation.

Author Response

Major corrections.

Comment 1: Methodology: the authors don´t mention BK loads quantification but present in results. Please add viral load procedures in methods.

What post-transplant stage are the patients in? 1 year? 6 months? What is the range?

Response 1: We agree with your comments and we added the viral load procedures for viruria, viremia in 2.2 section on materials and methods.

The average time of patients after transplantation is 28.26 months. (Table 2: Characteristics of kidney-transplant recipients).

Comment 2: Results: I missed an investigation into the possible impacts of the different genotypes and the mutations observed on variables such as viral load, detection of viremia and viruria, increase in creatinine, glomerular filtration and nephropathy, which would have been interesting to explore.

Response 2: We added the table of the comparison between BKV-I and BKV-IV as you suggested. The information was presented in the table 5. We also discussed about this problem with line 247-257.

Comment 3: Discussion: the cross-sectional nature of the investigation should, in addition to those presented above, have been also mentioned as an important limitation

Response 3: Thank you for your valuable comment. We added a limited cross-sectional study to the discussion (line 290-291).

Round 2

Reviewer 2 Report

All questions were fully replied and modified accordingly in the manuscript